# Role of Mitochondrial Transporters on Metabolic Rewiring of Pancreatic Adenocarcinoma: A Comprehensive Review

**DOI:** 10.3390/cancers15020411

**Published:** 2023-01-08

**Authors:** Graziantonio Lauria, Rosita Curcio, Paola Lunetti, Stefano Tiziani, Vincenzo Coppola, Vincenza Dolce, Giuseppe Fiermonte, Amer Ahmed

**Affiliations:** 1Department of Pharmacy, Health and Nutritional Sciences, University of Calabria, 87036 Rende, Italy; 2Department of Bioscience, Biotechnology and Environment, University of Bari, 70125 Bari, Italy; 3Department of Nutritional Sciences, The University of Texas at Austin, Austin, TX 78712, USA; 4Department of Pediatrics, Dell Medical School, The University of Texas at Austin, Austin, TX 78723, USA; 5Department of Oncology, Dell Medical School, LiveSTRONG Cancer Institutes, The University of Texas at Austin, Austin, TX 78723, USA; 6Department of Cancer Biology and Genetics, College of Medicine, The Ohio State University and Arthur G. James Comprehensive Cancer Center, Columbus, OH 43210, USA

**Keywords:** PDAC, metabolic rewiring, mitochondrial carriers, glutamine, aspartate

## Abstract

**Simple Summary:**

Metabolic rewiring is now considered a hallmark of cancer. Pancreatic ductal adenocarcinoma (PDAC) extensively rewires its metabolism especially for the utilization of glucose and glutamine that are mainly used to provide the biosynthetic intermediates for nucleotides, protein, and lipid synthesis and for redox homeostasis. In this regard, mitochondrial solute carriers play a key role in metabolic compartmentalization and hence metabolic rewiring. At least five mitochondrial transporters are involved in the metabolic reprogramming of PDAC. Targeting these transporters may provide an attractive strategy to combat this aggressive cancer. This review dissects the current knowledge about the role of these transporters in PDAC in order to explore the therapeutic potential of their targeting.

**Abstract:**

Pancreatic cancer is among the deadliest cancers worldwide and commonly presents as pancreatic ductal adenocarcinoma (PDAC). Metabolic reprogramming is a hallmark of PDAC. Glucose and glutamine metabolism are extensively rewired in order to fulfil both energetic and synthetic demands of this aggressive tumour and maintain favorable redox homeostasis. The mitochondrial pyruvate carrier (MPC), the glutamine carrier (SLC1A5_Var), the glutamate carrier (GC), the aspartate/glutamate carrier (AGC), and the uncoupling protein 2 (UCP2) have all been shown to influence PDAC cell growth and progression. The expression of MPC is downregulated in PDAC and its overexpression reduces cell growth rate, whereas the other four transporters are usually overexpressed and the loss of one or more of them renders PDAC cells unable to grow and proliferate by altering the levels of crucial metabolites such as aspartate. The aim of this review is to comprehensively evaluate the current experimental evidence about the function of these carriers in PDAC metabolic rewiring. Dissecting the precise role of these transporters in the context of the tumour microenvironment is necessary for targeted drug development.

## 1. Introduction

Pancreatic cancer is the fourteenth most common cancer and the seventh leading cause of cancer-associated death worldwide [1] and it is on the rise in Western countries. In the United States, for instance, pancreatic cancer is ranked as the fourth cause of cancer-related death and is predicted to be the second cause in 2030, thus overtaking breast and colorectal cancers [2,3]. The high lethality of PDAC is mostly due to late diagnosis, the lack of effective treatments, and the invariable occurrence of drug resistance. The late diagnosis is caused by the deferred onset of symptoms and the lack of specific biomarkers. On the other hand, the relatively low incidence of PDAC does not justify population screenings based on current technologies. In this scenario, more than 80% of patients are diagnosed with a locally advanced unresectable primary tumour or metastatic disease and surgery is not curative. Overall, pancreatic cancer patients have a poor 5-year survival of about 9% [4,5].

Pancreatic ductal adenocarcinoma (PDAC) is the most common form of pancreatic cancer accounting for more than 90% of diagnosed cases [6]. PDAC arises as a result of the malignant transformation of epithelial cells lining the ductal system that carries secretory enzymes and other substances away from the pancreas [7]. Genetically, PDAC is a complex malignancy involving aberrant chromosomal alterations, point mutations, and epigenetic modifications [8,9]. In this regard, oncogenic KRAS, and the tumour suppressors TP53, p16/CDKN2A, and SMAD4 are the most mutated genes in PDAC. KRAS is a small GTPase associated with cell surface receptors such as tyrosine kinase, which, once stimulated, activates various intracellular signalling pathways involved in cell proliferation, invasion, and metastasis [10]. KRAS is considered the major driver of PDAC and is mutated early in about 90% of PDAC cases, particularly in the elderly and in female patients [11].

Although the first observation of the metabolic changes in cancer highlighting that tumour cells consume more glucose to produce lactate and energy even in the presence of oxygen (known as the “Warburg effect”) was made a century ago, metabolic reprogramming in cancer has gained its importance only in the past two decades. Currently, metabolic reprogramming is considered a major hallmark of cancer [12] and it is investigated for cancer diagnosis, prognosis, and treatment [13,14]. PDAC is not an exception and its metabolism is known to be extensively reprogrammed in pathways utilizing glucose, lipids, and amino acids as nutrients [15]. For instance, PDAC cells markedly rewire their glucose metabolism in favour of increased glucose consumption via aerobic glycolysis and lactate production by upregulating both the glucose transporter (GLUT1) and glycolytic enzymes (such as phosphofructokinase 1 (PFK1), hexokinase 2 (HK2), and lactate dehydrogenase A (LDHA) (for reviews, see [16,17,18,19]) (Figure 1). Furthermore, another metabolic pathway tightly linked to glycolysis and markedly rewired in PDAC is the pentose phosphate pathway (PPP) [20]. Mutated KRAS activates non-oxidative PPP by induction of the ribulose-5-phosphate isomerase (RPIA) gene via the MAPK-MYC-RPIA pathway [21] (Figure 1). Moreover, PDAC cells markedly overexpressed monocarboxylate 1 and 4 transporters (MCT1 and MCT4, respectively), which are the two major lactate transporters, thus reducing the pH burden on glycolysis suppression and enabling NAD^+^ regeneration [22]. Lipid metabolism is also extensively reprogrammed in PDAC by upregulating enzymes involved in fatty acid synthesis including citrate synthase, ATP-citrate lyase (ACLY), acetyl-CoA carboxylase (ACC), and fatty acid synthase (for reviews see [23,24]). Cancer cells need high levels of lipid synthesis for membrane biogenesis and as signalling molecules or energy sources [25]. The uptake and catabolism of branched-chain amino acids (BCAAs) are significantly increased in PDAC cells, which, in turn, is linked to an increase in fatty acid synthesis. The lack of concomitant significant changes in mitochondrial metabolism, i.e., tricarboxylic acid (TCA) cycle intermediates and oxygen consumption rate, suggests that PDAC cells can use BCAA-derived acetyl-CoA as a precursor for fatty acid synthesis [26]. The metabolism of amino acids is also extensively rewired in PDAC. In particular, it is well demonstrated that PDAC cells use glutamine to replenish TCA cycle intermediates and provide nitrogen for the biosynthesis of purines, pyrimidines, non-essential amino acids, nicotinamide adenine dinucleotide (NAD), and glucosamine, as well as redox equivalents (NADPH) required for fatty acid synthesis and ROS scavenging [27,28] (Figure 1). Furthermore, PDAC takes up collagen from the tumour microenvironment through micropinocytosis, and collagen-derived amino acids are used as TCA intermediates to generate energy or building blocks useful for macromolecule synthesis, thus promoting PDAC survival under nutrient-limited conditions [29]. Likewise, the metabolic enzymes of one-carbon metabolism are upregulated in PDAC and associated with poor overall survival. These enzymes catalyze a series of metabolic reactions generating intermediates required for nucleotide synthesis and DNA methylation crucial for the cross talk of genetic and epigenetic alterations [30].

In the extensive metabolic reprogramming of PDAC cells, mitochondria play an active role, particularly in the rewiring involving glutamine and aspartate metabolism [31]. Glutamine enters the mitochondria where it is metabolized to replenish TCA cycle intermediates and to produce energy in the form of FADH_2_ and NADH and aspartate. The latter is transported to the cytosol where it is incorporated to proteins or participates in nucleotide synthesis and/or provides redox equivalents essential for fatty acid synthesis and ROS elimination [32]. However, these metabolites cannot diffuse freely across the inner mitochondrial membrane, and specific carriers are needed to enable their transport to and from the mitochondria. Mitochondrial transporters play a central role in the compartmentalization of these metabolic pathways [33]. Thus, mitochondrial transporters play crucial roles in the metabolic rewiring and many of them are found to be dysregulated in PDAC. In the present review, we aimed to critically and comprehensively gather the current knowledge about the role of five mitochondrial transporters known to be involved in the metabolic rewiring of PDAC. Readers interested in the role of plasma membrane transporters in pancreatic cancer are referred to published reviews [34,35].

## 2. The Glutamine Transporter SLC1A5_Var Is at the Centre of Glutaminolysis

Glutamine, the most abundant amino acid in human plasma, is used by cancer cells for versatile metabolic roles including the synthesis of proteins, nucleotides, lipids, and non-essential amino acids, or it acts as an exchanger for the transport of other amino acids [36,37]. Glutamine is also metabolized into glutamate, which enters the TCA cycle as α-ketoglutarate (α-KG) to be used for energy harvesting, generation of precursors for gluconeogenesis, and synthesis of antioxidants such as glutathione and NADPH [37]. Cancer cells show extreme dependence on glutamine for their growth and survival, a phenomenon known as “glutamine addiction” [38]. In particular, this holds true for PDAC cells, as the rewiring of their metabolism towards glutamine reliance is well documented. In this regard, pancreatic cancer cells deprived of glutamine drastically lose their ability of growth [28]. Glutamine metabolism in PDAC is initiated by the glutaminase (GLS), a critical enzyme that converts glutamine into glutamate, the first metabolic reaction of glutaminolysis [27,28]. Human GLS is encoded by two genes, *GLS* and *GLS2*, localized on chromosomes 2 and 12, respectively. *GLS* encodes two transcripts arising from alternative splicing: a long isoform (KGA; containing the 1–14 and 16–19 exons) found in the brain, and a short isoform (GAC; containing the first 15 exons) usually overexpressed in cancers [39]. Similarly, *GLS2* encodes two transcripts named GAB and LGA, arising from transcription driven by surrogate promoters. GLS2 is transcriptionally regulated by TP53 and it is hypermethylated in several cancers such as glioblastoma and hepatocellular carcinoma, suggesting that it is a tumour suppressor gene [40,41]. Indeed, forced expression of GLS2 into the glioblastoma cell lines U87MG and LN229 sensitizes them to the alkylating agent Temozolomide and H_2_O_2_-mediated oxidative stress by suppressing the PI3K/AKT pathway [42].

In PDAC, the GLS isoform GAC plays a key role in upregulating glutaminolysis and thus glutamine metabolic rewiring [28]. GLS seems to be regulated at transcriptional, post-transcriptional, and post-translational level. In this regard, Kim et al. found that the transcription factor EB (TFEB) could directly bind to the GLS promoter and upregulate its expression in PDAC cell lines. TFEB knockdown resulted in the downregulation of GLS expression and glutamine metabolism and consequent tumour growth suppression both in vitro and in xenograft models [43]. GLS is also post-transcriptionally regulated by increasing mRNA stability, for instance, by repressing the expression of miR-23a/b (an inhibitor of GAC) the oncogenic MYC upregulates its expression [44]. Interestingly, a recent study in PDAC showed that GAC is post-translationally regulated through succinylation. In this regard, P38 mitogen activated protein kinase (MAPK) phosphorylates succinyl-CoA ligase [ADP-forming] subunit β (SUCLA2) at ser-79, leading to GLS-SUCLA2 dissociation, which in turn enhances GLS K311 succinylation and ultimately increases its activity [45].

Despite reports showing that GAC is localized in the inner mitochondria membrane, the exact localization of its catalytic domain has not yet been elucidated [46,47,48]. Clarifying this aspect is of paramount importance because if the catalytic site faces the intermembrane space, then a glutamate transporter will be required. On the contrary, if the catalytic side faces the matrix, then a glutamine transporter would be needed [49] (Figure 2). If the former holds true, it would be critical to investigate whether the glutamate enter the matrix via AGC or GC, which can both efficiently perform the task. However, these two transporters are different regarding the export of aspartate. In essence, if the glutamate entered through AGC, then the aspartate produced in the matrix could exit through this transporter in exchange for glutamate. On the other hand, if glutamate entered through GC, then an aspartate exporter, such as UCP2, is needed. Evidence in PDAC shows that glutamine is transported via SLC1A5_Var from the cytosol to matrix (see below), and aspartate is transported via UCP2 that is also known to be overexpressed in PDAC cells (discussed later).

The search for the mitochondrial glutamine transporter began in the 1970s [50]. However, only recently has light been shed on this transporter. In this regard, Yoo and colleagues showed that the gene encoding the plasma membrane sodium-dependent neutral amino acid transporter SLC1A5 (also designated as ASCT2) has two different transcription initiation sites. The second transcription site occurs in the first intron, and results in a shorter transcript lacking exon 1 and encoding a 339 amino acid protein therein defined as SLC1A5_Var that is localized in the inner mitochondrial membrane and seems to be responsible for the influx of glutamine towards the matrix [51].

This transporter, similarly to the mitochondrial pyruvate carrier (MPC) [52,53], lacks a tripartite structure characterizing the mitochondrial carrier family SLC25 [54,55,56,57]. SLC1A5_Var is overexpressed in several adenocarcinomas and its increased levels were reported to be associated with poor prognosis [51]. Hypoxia can induce the expression of SLC1A5_Var via HIF-2α. A functional analysis involving knockdown and overexpression showed that SLC1A5_Var plays an active role in the upregulation of glutaminolysis favouring improved redox homeostasis and the induction of gemcitabine resistance in pancreatic cancer, thus suggesting an oncogenic role for SLC1A5_Var [27,51]. This is crucial because PDAC is generally characterized by severe hypoxia arising from the dense desmoplastic stroma, poor vascularization, and high proliferation rate of cancerous cells, leading to an imbalance between oxygen consumption and supply [58]. Accordingly, once transported into the mitochondria, glutamine is converted into glutamate by glutaminase. Glutamine-derived glutamate is further converted into aspartate and α-KG by the action of the mitochondrial glutamate-oxaloacetate transaminase 2 (GOT2) [28]. Then, aspartate is transported to the cytosol, where it is incorporated into proteins, and used for nucleotide synthesis, but also for the generation of NADPH reducing equivalents for ROS scavenging and redox homeostasis [59,60] (Figure 2). Conversely, SLC1A5_Var silencing can result in increased ROS level, and decreased glutaminolysis and GSH/GSSG ratio [58]. Unfortunately, the study where this finding was reported did not measure the levels of cytosolic aspartate. However, a reduction is expected upon SLC1A5_Var knockdown, as a positive association between glutaminolysis and cytosolic aspartate levels has been reported by other groups [28,32,61]. It should be noted here that concerns have been raised regarding this variant as a mitochondrial glutamine carrier [62]. These concerns are in regards to the method used for mitochondrial signal peptide prediction, to the antibodies used for its detection, as well as to the lack of the first 203 amino acids forming four transmembrane helices of the SLC1A5, which are predicted to be crucial for the interaction with lipid bilayers, thus questioning the capability of this N-terminal truncated SLC1A5_Var to act as a transporter [62]. Hence, further experimental evidence confirming its mitochondrial localization and its transport function by in vitro assays in liposomes is urgently needed before any potential of SLC1A5_Var as a therapeutic target can be considered.

## 3. Mitochondrial Glutamate Carriers: SLC25A22 and SLC25A18

The glutamate present in the cytosol obtained either from diet or protein degradation and/or from the interconversion of other amino acids is transported into mitochondria by glutamate carriers (GC). In humans as in other organisms, two different isoforms are known, GC1 and GC2, encoded by the *SLC25A22* and *SLC25A18* genes, respectively [63,64]. Both carriers catalyze glutamate–proton symport into the matrix, but the two proteins show different tissue expression levels and kinetics properties [63,65] (Figure 2). In KRAS-mutated PDAC, glutamine-derived glutamate appears to bypass the need for a mitochondrial glutamate carrier, as supported by the discovery of the glutamine carrier described above [51] (Figure 2). However, this notion should be treated with caution. First, GC1 is known to be highly expressed in the pancreas at the mRNA and protein levels [63,66]. Second, in KRAS-mutated colorectal cancer (CRC), SLC25A22 was shown to be essential for CRC cell growth in glutamine-containing media and in xenograft models [67]. In this regard, Wong et al. showed that SLC25A22 expression is upregulated in KRAS-mutated CRC cell lines and tumour tissue compared to wild-type KRAS cell lines and adjacent non-tumour tissues, respectively [67]. SLC25A22 silencing in KRAS-mutated CRC cell lines reduces glutamine metabolism, as revealed by a reduction in TCA cycle intermediates including succinate, fumarate, malate, oxaloacetate, and aspartate, and this was rescued by aspartate supplementation [67]. SLC25A22 glutamate transport function seems crucial for cell proliferation, in vitro migration, and invasion and for tumour metastasis in xenograft models, and it was associated with poor prognosis in patients [67]. Glutamate-derived aspartate enhances glucose metabolism via regeneration of NAD^+^, and ameliorates redox homeostasis in favour of cell growth, migration, and invasion [67]. Third, knocking down SLC25A22 in the KRAS-mutated PDAC cell line SW1990 was shown to suppress colony formation and raises questions about the dispensability of the glutamate carrier in PDAC cells as a result of SLC1A5_Var glutamine carrier expression [67]. Finally, GC1 is also overexpressed in osteosarcoma and gallbladder cancer (GBC) [68,69]. Gain of function (overexpression) of GC1 resulted in promotion of GBC cell lines proliferation as well as tumour growth and metastasis in xenograft models. Conversely, GC1 silencing induced GBC cell apoptosis by the downregulation of Bcl-2 and by the upregulation of cleaved PARP, cytochrome-c, and BAX mediated by the MAPK/ERK pathway [69]. In osteosarcoma, overexpression of SLC25A22 increased osteosarcoma cells proliferation, invasion, and migration in vitro, as well as tumour growth and lung metastasis of in vivo xenograft models. Furthermore, SLC25A22 expression was associated with poor patient survival. SLC25A22 was associated with a reduced expression of phosphatase and tensin homolog (PTEN) and an increased phosphorylation of protein kinase b (Akt) and Focal Adhesion Kinase (FAK) [68].

SLC25A18 is poorly studied in the context of cancer, including pancreatic cancer. Interestingly, SLC25A18 is downregulated in colorectal cancer, and its overexpression reduces the Warburg effect and cell proliferation, and it was associated with a longer disease-free survival time. SLC25A18 suppressed the expression of CTNNB1, PKM2, LDHA, and MYC indicating that its tumour suppressive effect might be mediated by the inhibition of the Wnt/β-catenin pathway [70]. These studies indicate that there are open questions about the role of GC1 and GC2 in PDAC, which deserve further investigation.

## 4. The Aspartate–Glutamate Carriers SLC25A12 and SLC25A13

The aspartate–glutamate carriers AGC1 and AGC2 can also transport glutamate into the matrix. They belong to the mitochondrial carrier family SLC25 and are transcribed from two different nuclear genes, namely *SLC25A12* (also called aralar1) and *SLC25A13* (also called citrin) [71,72]. Both aralar1 and citrin exchange in a Ca^2+^-dependent manner mitochondrial aspartate for cytosolic glutamate (which may also be generated in the intermembrane space) plus one proton [73] (Figure 2). The Ca^2+^-dependent stimulation of both aralar1 and citrin is enabled by the presence of a unique long N-terminal extension harbouring a number of EF-hand motifs and facing the intermembrane space [74]. Although AGC1 and AGC2 catalyze the same transport, they show specific tissue expression: AGC1 is mainly expressed in the heart, brain, and skeletal muscles, while AGC2 is expressed in the liver, gallbladder, and gastrointestinal tract, among others [73,75]. Together with the malate/α-ketoglutarate carrier (also called 2-oxoglutarate carrier, OGC), AGC1 and AGC2 form the malate–aspartate shuttle, which is responsible for shuttling NADH reducing equivalents from the cytosol to mitochondria [76,77].

Although the dysregulated function of AGC1 and AGC2 has been widely investigated in the context of inborn errors (for more details, see [78,79,80,81]), their role in cancer is scarcely known. In this regard, AGC2 was found to be overexpressed in melanoma cancer cell lines and is associated with enhanced cell proliferation, invasion, and poor prognosis [82,83]. AGC2 is also overexpressed in CRC cell lines under glucose-depleted conditions and associated with increased tumour aggressiveness and poor prognosis [84]. As reported by Alkan et al., AGC1 silencing reduces the proliferation rate of C2C12 cells as well as LLC1 and H1299 (lung), PANC1 and Capan2 (pancreatic), and Hela (cervical) cancer cells, due to impaired aspartate metabolism [85]. This was particularly evident in glutamine-depleted media and was reversed by aspartate supplementation. In glutamine-replete media, AGC1 silencing did not block cell proliferation, suggesting the existence of another transporter that allows aspartate to exit mitochondria. In fact, AGC1 knockdown potentiated the effects of the mitochondria glutaminase inhibitor CB-839 in vivo. This is consistent with a model where AGC1 is involved in mitochondrial aspartate export under glutamine-limiting conditions, while an alternative mitochondrial carrier characterized by a higher Km value for aspartate replaces AGC1 under high glutamine concentration [85,86]. In this regard, the uncoupling protein 2 (UCP2) present in the inner mitochondrial membrane is capable of transporting mitochondrial aspartate in exchange for phosphate plus a proton [87] (Figure 2). UCP2 has a Km value for aspartate of 6.84 mM, which is about 100-fold higher than that of AGC that is about 50 µM [73]. A similar mechanism could be at play in regard to the entry of glutamate into mitochondria of CRC and PDAC cells mediated by GC1 as discussed above [67]. GC1 has a Km value for glutamate of 5.18 mM [63], which is 30-fold higher than that of AGCs. Overall, evidence suggests that in cancers with highly rewired glutamine metabolism such as PDAC, the flux of aspartate and glutamate occurs via mitochondrial carriers having a high Km value, which are not easily saturable. Indeed, GC1 and UCP2 (discussed below) have been shown to play a key role in the glutamine metabolic rewiring in pancreatic cancer cells (Figure 2). However, under glutamine-depleted conditions, AGC1 may play a key role in this transport owing to its low Km values (Figure 2).

## 5. UCP2: A Mitochondrial Aspartate Transporter

UCP2 is the most commonly investigated mitochondrial carrier in cancer and specifically in PDAC [88,89,90,91]. This transporter is a member of a subfamily of the SLC25 mitochondrial family known as uncoupling proteins and is thought to allow protons to enter the mitochondrial matrix, thereby dissipating the proton gradient generated by the activity of the respiratory electron transport chain, thus uncoupling respiration from ADP phosphorylation [92]. To date, six human isoforms (UCP1–6) encoded by six different genes have been identified, namely *SLC25A7*-*9*, *SLC25A27, SLC25A14,* and *SLC25A30* [93]. All UCPs share the amino acid sequences called “uncoupling protein signatures”, which are positioned in the first, second, and fourth helices, as well as in the second matrix loop. The six UCPs also share high homology in the purine nucleotide binding domain [94]. UCP1 was the first member to be identified of the UCP subfamily [95]. All the others were subsequently discovered based on their sequence homology with UCP1 [96]. The uncoupling capabilities of UCP2–5 appear to be modest. Their mild mitochondrial uncoupling could prevent excessive mitochondrial reactive oxygen species (ROS) production, thereby decreasing cell oxidative damage [97], as recently demonstrated for UCP2 and UCP3 in stroke and ischemia/reperfusion [98].

Human UCP2 is encoded by the *SLC25A8* gene mapped to chromosome 11, and it has 9 exons and 8 introns [99]. It was first reported in 1997 by Fleury et al. [100] and described as an uncoupling protein or UCP2 [100] because of its high identity (59% amino acids) with UCP1. UCP2 is ubiquitously expressed in the liver, gallbladder, pancreas, spleen, gastrointestinal tract, adipose tissue, and immune cells [101,102]. As a result of this wide tissue distribution, its thermogenic function in the adipose tissue was put into question. Indeed, later studies involving animal models ruled out its function in non-shivering thermogenesis, they instead revealed a role for UCP2 as an antioxidant protein/modulator involved in redox homeostasis [103]. UCP2 is tightly regulated at transcriptional, translational, and post-translational level [104,105]. At the transcriptional level, UCP2 is negatively regulated by the TGF-β signalling pathway via the tumour suppressor protein SMAD4 [106]. SMAD4 is inactivated in several types of cancers, particularly in more than 50% of PDAC leading to an increased expression of UCP2 [107,108]. Glutamine, but not its downstream metabolites, upregulates UCP2 translation. On the contrary, glutamine depletion results in a rapid decline in UCP2 protein levels [109,110]. In this regard, UCP2 has a very short half-life as it is rapidly targeted for proteasomal degradation [111]. UCP2 expression is regulated by miRNAs that include miR-133a [112,113] and miR-15a [114], both considered as tumour suppressor miRNAs downregulated in PDAC [115,116]. In addition, miRNA-214 upregulates UCP2 expression ameliorating the oxidative stress associated with diabetic nephropathy [117]. miRNA-214 is considered to be an oncomiRNA upregulated in cancers such as PDAC [118]. Mechanistically, it has been reported that miR-2909 can upregulate UCP2 expression by repressing the KLF4 gene, which is a tumour-suppressing gene downregulated in PDAC [119,120]. Moreover, UCP2 is negatively regulated post-translationally by glutathionylation [121,122]. In this context, reduced ROS levels stimulate UCP2 glutathionylation, and thus its inactivation. To counteract the increase in ROS levels, UCP2 is deglutathionylated and the active form is released [123].

The first evidence for UCP2’s unknown role as an amino acid carrier was provided by Vozza et al. [87]. UCP2, functionally reconstituted into liposomes, was able to transport aspartate, malate, and oxaloacetate in exchange for phosphate plus a proton [87]. In line with this function, UCP2 silencing in hepatocellular carcinoma (HepG2) increased the mitochondrial membrane potential and the ATP/ADP ratio while it decreased lactate production when cells were grown in the presence of glucose. Interestingly, the opposite effect was observed in the presence of glutamine [87]. In the presence of both glutamine and glucose, the mitochondrial level of TCA cycle intermediates (citrate, malate, fumarate, 2-oxoglutarate) was increased [87]. This is consistent with a cataplerotic function in which UCP2 pumps TCA cycle intermediates to the cytosol, thereby disrupting the complete oxidation of glucose by the TCA (due to oxaloacetate depletion), while rewiring metabolism towards enhanced glutaminolysis. This cataplerotic function of UCP2 is of crucial importance in cancer metabolism because many cancers are known to rewire their metabolism towards enhanced glutamine utilization or “glutamine addiction”. In line with this notion, UCP2 overexpression has been reported in several cancers and is associated with chemoresistance [61,89,124,125,126,127]. Traditionally, the tumour-promoting effect of UCP2 has been ascribed to a reduction in ROS levels in cancer cells, achieved by lowering the electrochemical gradient across the inner mitochondrial membrane due to its protonophoric activity. However, this theory has been challenged by several authors who reported that UCP2 lacks uncoupling activity [128,129]. In line with this conclusion, we have recently showed that in KRAS-mutated PDAC cancer cell lines, UCP2 expression decreases ROS levels by exporting aspartate out of mitochondria [32]. In comparison with non-silenced controls, the silencing of UCP2 reduces glutaminolysis, lowers both the NADPH/NADP^+^ and the GSH/GSSG ratios and increases the levels of ROS. In addition, UCP2 silencing significantly reduced tumour size of KRAS-mutant PDAC xenografts [32]. Notably, the silencing of UCP2 reduces glutaminolysis in both KRAS-mutant and KRAS wild-type cells, but only reduces the cell proliferation and tumour growth of KRAS-mutant cells [32]. This result suggests that UCP2 plays a key role in glutaminolysis and that the increased levels of ROS observed upon UCP2 silencing in KRAS-mutated PDAC cells are most likely produced by impaired glutaminolysis rather than reduced UCP2-dependent uncoupling activity [32] (Figure 2). Moreover, UCP2 expression does not appear to be regulated by mutated KRAS. In fact, PDAC BxPC3 cells normally carrying the wild-type KRAS do not alter their UCP2 levels when overexpressing the KRAS G12V mutant, but rather show increased proliferation and colony formation capacity. On the other hand, these effects are significantly reduced by UCP2 silencing, thus reinforcing the conclusion that UCP2-dependent aspartate transport is critical to support the increase in cell proliferation induced by mutated KRAS [32].

It should be emphasized here that, in PDAC, mutated KRAS induces rewiring of the pentose phosphate pathway by uncoupling oxidative from non-oxidative reactions and limits it to the mere production of ribose-5-phosphate for nucleotide synthesis [19,130]. In order to fulfil the increased demand for NADPH, KRAS also rewires glutamine metabolism by raising the expression of both glutaminase and GOT2. These enzymes metabolize glutamine into α-KG and aspartate: the former is expended through the TCA cycle for energy production, and the latter is transported out of mitochondria into the cytosol, where it is used for the synthesis of proteins and nucleotides and/or is converted by GOT1 into oxaloacetate and malate. Finally, malate is converted to pyruvate by the malic enzyme, a reaction accompanied by NADPH production [28,131,132] (Figure 2). It should also be stressed again that UCP2 has a high Km value for aspartate (6.8 mM), suggesting that UCP2 might be involved in aspartate transport only under conditions of active glutaminolysis, such as in PDAC [87]. In this context, the mitochondrial glutamine carrier (i.e., SLC1A5_Var) could transport glutamine to the matrix, then the glutamine-derived aspartate is transported out through UCP2 (Figure 2). This metabolic model is supported by the fact that the silencing SLC1A5_Var or UCP2 was shown to cause reduced PDAC cell growth [32,51] and this effect was rescued by aspartate or glutamate and in the presence of glutamine, both transporters are critical to fulfil the cytosolic need for aspartate [32,51]. Conversely, when glutaminolysis is impaired or glutamine is limited, glutamate can enter the matrix likely through AGC and produces the aspartate that is able to exit via the same transporter.

Most of the published data on the role of UCP2 in PDAC have been obtained using established cancer cell lines and animal models, which means that researchers have only investigated the role of UCP2 in PDAC maintenance and progression, whereas the role of UCP2 on PDAC initiation remains largely unexplored. Recently, a study published by Aguilar et al. demonstrated that UCP2 was overexpressed only on established murine colorectal tumours and human patients compared to their normal counterpart tissues. Interestingly, knocking out UCP2 enhances tumourigenesis of the colon and small intestine in a carcinogen-induced and Apc^-/-^ mouse model, respectively. These data suggest that UCP2 might act as a tumour suppressor during initiation [133]. In this regard, the study showed that UCP2 deletion induced a metabolic reprogramming in which glucose-derived pyruvate was directed to fatty acid synthesis by increasing the synthesis of citrate, a metabolic change that requires high amounts of NADPH [133]. In parallel, UCP2-KO CRC cells showed a decrease in glucose-6-dehydrogenase activity, and thus in the GSH/GSSG ratio, as well as increased ROS levels [133]. It was speculated that the oxidative stress generated in this context could cause genomic instability and hence tumour initiation [133]. This hypothesis is also consistent with the finding that UCP2 deletion suppressed C4 metabolites efflux from the mitochondrial matrix, thereby promoting the synthesis of citrate from acetyl-CoA and oxaloacetate. Similar results were also reported using HepG2 cells, neonatal rat ventricular myocytes, and human pluripotent stem cells [87,134,135].

In summary, evidence from the literature suggests that UCP2 may act as a tumour suppressor during tumour initiation while it can have the opposite role as tumour promoter during progression, though exerting the same biochemical function at both stages [136,137].

## 6. The Mitochondrial Pyruvate Carrier

Normally, glucose is utilized by mammalian cells as the main energy source where it is first metabolized through a ten-reaction pathway, glycolysis, which results in two molecules of pyruvate and the net energy of 2 ATP molecules [138]. Next, pyruvate is transported to the mitochondria to be further metabolized through the TCA cycle generating the remaining ATP molecules. For this reason, pyruvate is at the crossroad between glycolysis and mitochondrial oxidative phosphorylation [139]. Pyruvate, like many cytosolic metabolites, cannot bypass the inner mitochondrial membrane. The existence of a specific mitochondrial pyruvate carrier was hypothesized in the 1970s. However, the molecular identity of the molecular complex acting as pyruvate carrier was independently elucidated by two research groups only in 2012 [52,53].

The mitochondrial pyruvate carrier (MPC) is highly conserved throughout evolution from yeast to humans [140]. In humans, there are two genes encoding for MPC. MPC1 is encoded by the *MPC1* gene (also known as Brp44L) that maps to chromosome 6 and it has a transcript made of 8 exons and 7 introns giving rise to a 109 amino acid-long protein. MPC2 is encoded by the *MPC2* gene (also known as Brp44) mapped to chromosome 1 and it has five exons and four introns producing a 127 amino acid-long protein [52,53].

Unlike other known canonical mitochondrial carriers such as SLC25, MPC belongs to a new class of mitochondrial carriers, SLC54 [62]. Although the molecular weight of MPC1 and MPC2 is 12 and 15 KDa, respectively, the whole MPC complex has a molecular weight of 150 KDa, suggesting that MPC1 and MPC2 form hetero-oligomers with 1:1 ratio. The loss of either one of the two results in a carrier unable to uptake pyruvate [141]. Interestingly, it has recently been suggested that MCP2 could be able to function as an independent carrier of pyruvate [142].

Cancer cells dramatically reprogram glucose metabolism through increased glucose flux via the glycolytic pathway and lactate efflux to the extracellular microenvironment, as well as decreased pyruvate influx to mitochondrial oxidative phosphorylation even in the presence of oxygen and functional mitochondria [17,143]. The latter is achieved by (i) decreased pyruvate synthesis by downregulating the expression of the most active form of pyruvate kinase, which catalyzes the conversion of phosphoenolpyruvate (PEP) to pyruvate (for review, see [144,145]); (ii) downregulation of the MCP complex, particularly MPC1 [146,147]; and iii) inhibition of the pyruvate dehydrogenase complex (PDH) by increasing the expression of pyruvate dehydrogenase kinase, which phosphorylates PDH and reduces its activity (for review see [148,149]). Under this condition of decreased pyruvate utilization and accumulation of its precursors (glycolytic intermediates), cancer cells switch their glycolytic metabolism to three main biosynthetic pathways: (i) the pentose phosphate pathway in order to synthesize ribose-5-phosphate for nucleotides synthesis; (ii) the serine synthesis pathway to be used in one-carbon metabolism and involved in nucleotide synthesis; and (iii) the hexosamine pathway in order to synthesize amino sugars for the biosynthesis of glycoproteins and glycoconjugates [150]. Therefore, the MCP complex plays a key role in glucose metabolic rewiring and cancer pathogenesis. As recognized soon after its discovery, the MPC complex, and MPC1 in particular, is downregulated in many cancers [149,151,152,153,154]. In PDAC, evidence of the MPC complex downregulation is scarce and consists of only two reports published to date. Cui and colleagues noted a significant reduction in MPC1 expression (both at the mRNA and protein levels) in PDAC cell lines and tumour tissues compared to adjacent non-tumour controls. This was associated with poorer differentiation, lymph node metastasis, higher TNM (tumour (T), nodes (N), and metastases (M)) stages, and patients’ overall survival [155]. Moreover, MPC1 downregulation was accompanied by a decrease in mitochondrial pyruvate and an increase in cell growth, invasion, migration, tumourigenicity, and lactate production (Figure 2). These effects were reversed by MPC1 overexpression [155]. Mechanistically, the reduction in MPC1 expression appears to be mediated by the lysine demethylase 5A (KDM5A) that binds to the MPC1 promoter and demethylates H3K4 thereby suppressing MPC1 expression. Evidence supporting this mechanism is provided by experiments knocking down KDM5A that causes an increase in MPC1 expression and reverses all parameters associated with its downregulation and by the fact that when KDM5A is overexpressed in PDAC, it promotes cell proliferation in vitro and tumour growth in vivo by suppressing MPC1 expression. These data suggest that the KDM5A/MPC-1 signalling pathway increases PDAC growth, invasion, and migration by decreasing pyruvate metabolism, thus highlighting it as a hotspot for targeted drug development [155]. Indeed, KDM5A has been extensively investigated as a drug target for cancer therapy (see review [156]). Another report showed that MPC1 is downregulated in pancreatic cancer cells, and further knockdown of MPC1 resulted in a spindle-like shape, in expression changes of epithelial–mesenchymal transition (EMT) markers such as E-cadherin and fibronectin, as well as in the acquisition of a migratory phenotype. These alterations were accompanied by an increase in GLS expression and significantly increased resistance to radiotherapy. Unfortunately, this study did not investigate the mechanism linking MCP1 downregulation and/or GLS overexpression to EMT [157].

## 7. Conclusions and Perspectives

PDAC is characterized by extensive metabolic rewiring including enhanced aerobic glycolysis and glutaminolysis or glutamine addiction. In this context, glucose is metabolized through aerobic glycolysis for rapid energy production and to provide the building blocks for biosynthetic pathways including the pentose phosphate pathway, hexosamine pathway, and serine biosynthesis (Figure 1). On the other hand, glutamine is directly used to synthesize nucleotides, hexosamines, and proteins, as well as to produce energy through the TCA cycle (Figure 1). Additionally, glutamine can be converted into metabolites such as aspartate, which in turn is involved in the biosynthesis of nucleotides and asparagine, incorporated into proteins, and provides NADPH redox equivalents for maintaining redox balance (Figure 1). In order to produce aspartate, glutamine must be transported to the mitochondrial matrix, where it is converted into aspartate and α-KG by GLS and GOT2, two key enzymes of glutaminolysis.

To favour metabolic rewiring, PDAC cells change the transport of metabolites through the inner mitochondrial membrane towards the matrix by decreasing the flux of pyruvate, increasing that of glutamine and glutamate, while increasing the efflux of aspartate. Hence, the mitochondrial solute transporters MPC, SLC1A5_var, SLC25A12, SLC25A22, and UCP2 play a central role in these metabolic reprogramming scenarios (Figure 2). Data obtained in experimental settings with the gain and loss of function of these transporters have suggested that they have a main role in PDAC progression and maintenance [32,51,67,155]. However, the current literature suggests that there might be additional mitochondrial transporters involved in PDAC progression such as SLC25A44 (mitochondrial branched-chain amino acids (BCAA) transporter) [158] and SLC25A1 (mitochondrial citrate transporter), both of which should be further investigated [159,160,161]. Indeed, a study by Lee et al. demonstrated that PDAC cells increase the uptake of BCAAs that are further metabolized in the matrix by branched-chain amino acid transaminase 2 (BCAT2; the mitochondrial isoform) and BCKDH, thus resulting in enhanced fatty acid synthesis [26]. According to this scenario, it is reasonable to hypothesize that the expression of the mitochondrial BCAA transporter and of the mitochondrial citrate carrier would also increase the flux of BCAAs and efflux of citrate to the cytosol where fatty acid biosynthesis takes place.

From a patient treatment perspective, PDAC metabolic rewiring offers new attractive therapeutic targets. To date, most efforts have been put into the design and development of potent inhibitors of glucose and glutamine metabolism, such as those targeting glutaminase (CB-839) and hexokinase (Benitrobenrazide), for example. On the other hand, studies aimed at the treatment of PDAC targeting the mitochondrial carriers are very limited, likely because of the very recent discovery of their role in the pathogenesis of the disease and the challenges and limitations intrinsic to in vitro inhibition assays. However, preliminary evidence shows that targeting mitochondrial carries with small molecules, particularly those of natural origin, could provide powerful tools to fight PDAC [162]. For instance, the UCP2 inhibitor genipin (chemical compound extracted from the *Genipa americana* fruit) has shown promising effects against PDAC cell lines by inducing GAPDH nuclear translocation and autophagy [163], increasing ROS productions and gemcitabine sensitization [125], and reducing the expression of hnRNPA2/B1, a key regulator of GLUT1, PKM2 mRNAs, and lactate dehydrogenase (LDH), thereby sensitizing PDAC cell lines to glycolysis inhibition by 2-deoxy-D-glucose [164]. Therefore, the identification of additional natural compounds targeting mitochondrial carriers should be further explored. However, small molecule-based therapy is associated with a number of pitfalls including, but not limited to, the cost and time of the screening of libraries to find effective lead molecules, the potential off-targets, the toxicities and the side effects that may arise during preclinical and clinical trials, and the insurgence of drug resistance [165]. Finally, the use of miRNAs that specifically target the transcript of the molecule of interest by inhibiting its translation or inducing its degradation could represent a valid alternative strategy to overcome the challenges associated with small molecules. In this regard, the expression of UCP2 can be downregulated by miR-15a and miR-133a, two miRNAs known to be downregulated in PDAC [166,167]. Further studies are needed to explore the effects of such miRNAs on their targets and other transporters.

## Figures and Tables

**Figure 1 cancers-15-00411-f001:**
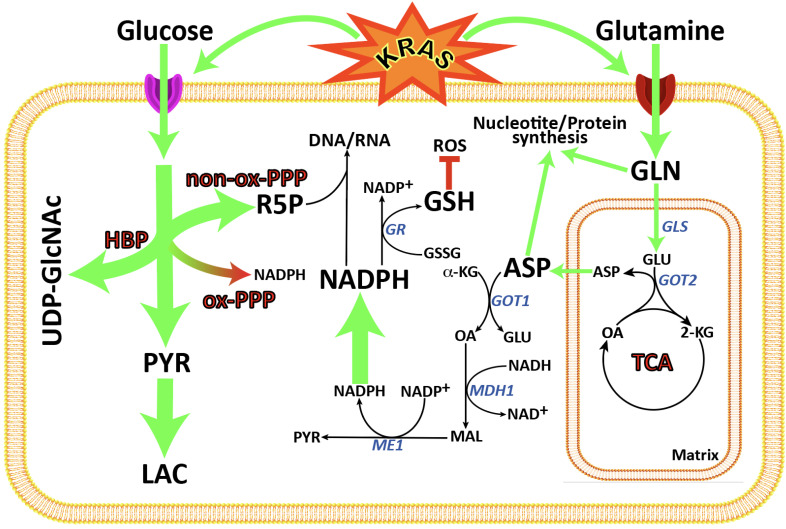
A synopsis of metabolic pathways rewiring in KRAS-mutant PDAC cell lines. KRAS enhances both glucose and glutamine uptake in PDAC cell lines. Glucose is metabolized merely through aerobic glycolysis to produce intermediates used for pentose phosphate and hexosamine pathways as well as lactate. Glutamine can directly be used for protein and nucleotide synthesis or fluxed into mitochondria where it is converted to aspartate by the action of glutaminase and GOT2. Produced aspartate is transported to the cytosol where it is incorporated into protein or used for synthesis of nonessential amino acids and nucleotides. PDAC cells use aspartate for the generation of NADPH and reduced glutathione by action of GOT1, MDH1, and ME1 to control the redox homeostasis. HBP, hexosamine biosynthetic pathway; PPP, pentose phosphate pathway; TCA, Krebs cycle; GLS, glutaminase; GOT1, glutamate-oxaloacetate transaminase 1; GOT2, glutamate-oxaloacetate transaminase 2; GR, glutathione reductase; MDH1, cytosolic malate dehydrogenase; ME1, cytosolic malic enzyme; α-KG, α-ketogluturate; Asp, aspartate; Glu, glutamate; Gln, glutamine; GSH, reduced glutathione; GSSG, oxidized glutathione; Lac, lactate; Mal, malate; OA, oxaloacetate; Pyr, pyruvate; R5B, ribose-5-phosphate; ROS, reactive oxygen species; UDP-GlcNAc, Uridine diphosphate N-acetylglucosamine.

**Figure 2 cancers-15-00411-f002:**
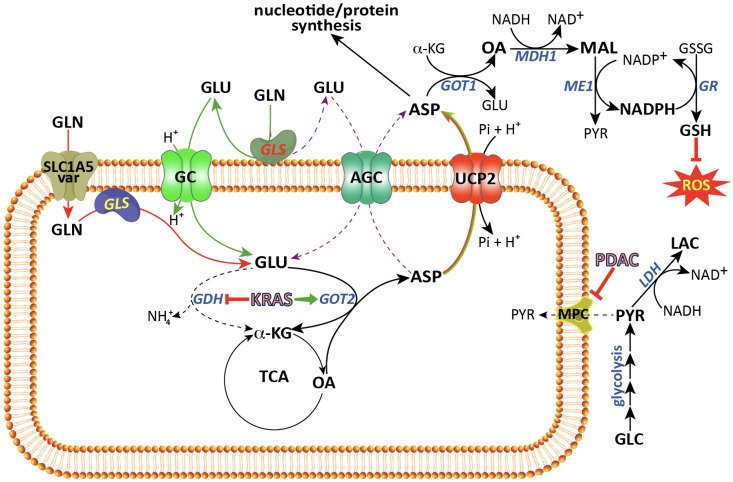
Role of mitochondrial solute carriers in PDAC metabolic rewiring. Depending on the localization of GLS either in the matrix or in the intermembrane space, two scenarios are proposed. In the first (red lines), glutamine is transported to the matrix through SLC1A5_Var and converted by the action of GLS localized in the matrix to glutamate which in turn is metabolized to aspartate and transported out of mitochondria by uncoupling protein 2 (UCP2). In the second (green and violet lines), glutamine is converted to glutamate by GLS localized in the intermembrane space and glutamate thus produced is transported to the matrix by glutamate carrier (GC) (green lines) or aspartate–glutamate carrier (AGC) (violet dashed lines) and glutamate-derived aspartate exists the mitochondria through UCP2 (green lines) or AGC itself (violet dashed lines), respectively. In the cytosol, aspartate reacts with α-KG by GOT1 to produce glutamate and oxaloacetate. The latter is reduced into malate by the action of MDH1 and malate is converted by ME1 to pyruvate and NADPH used for GSSG reduction and ROS control. Finally, in PDAC, mitochondrial pyruvate carrier (MPC) is downregulated hence reducing glucose flux towards OXPHOS and increases lactate production. TCA, Krebs cycle; AGC, aspartate–glutamate carrier; GLS, glutaminase; GC, glutamate carrier; GDH, glutamate dehydrogenase; GOT1, cytosolic glutamate-oxaloacetate transaminase; GOT2, matrix glutamate-oxaloacetate transaminase; GR, glutathione reductase; LDH1, lactate dehydrogenase; MDH1, cytosolic malate dehydrogenase; ME1, cytosolic malic enzyme; MPC, mitochondrial pyruvate carrier; UCP2, uncoupling protein 2; α-KG, α-ketoglutarate; Asp, aspartate; Glc, glucose; Glu, glutamate; Gln, glutamine; GSH, reduced glutathione; GSSG, oxidized glutathione; Lac, lactate; Mal, malate; Pyr, pyruvate; OA, oxaloacetate; ROS, reactive oxygen species.

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
