# Peer review of "Role of Mitochondrial Transporters on Metabolic Rewiring of Pancreatic Adenocarcinoma: A Comprehensive Review"

_cancers, 2023, doi:10.3390/cancers15020411_

Round 1

Reviewer 1 Report

The study by G.Lauria et al. is a comprehensive review of metabolic reprogramming as a major mechanism of pathogenesis of pancreatic ductal adenocarcinoma (PDAC).  The authors presented an in-depth, high quality analysis of the most important metabolic pathways involved in the biology of this clinically aggressive malignancy.  From the biochemical viewpoint the study is indeed excellent. 

Nevertheless, given that the audience of Cancers is not limited to the experimentalists, I suggest the authors focus on specific oncological problems. In particular, what is the role of individual metabolic changes in the natural history of PDAC, namely, is there a difference between early and late stages of the disease, as well as between the primary tumor vs metastases? Which biochemical events can serve as clinical markers of diagnosis and/or prognosis in different histological subtypes? Is there a correlation between the expression of KRAS and the degree of rewiring, or these events may be mutually dependent? Also, a section about targeting metabolic mechanisms with small molecules is worthy if the literature analysis allows to generate therapeutically relevant hypotheses. 

I believe these considerations will make the otherwise very good review perfectly appropriate for the oncological journal. 

Author Response

We appreciate the reviewer’s raised consideration. However, the review aimed to dissecting the role of mitochondrial carriers in metabolic rewiring with particular focus on glutamine, glutamate, and aspartate transporters as well as mitochondrial pyruvate carriers and our brief discussion on metabolic rewiring was introductory to help the readers appreciating the metabolic importance of these carriers. Furthermore, as also stressed in several points of the manuscript, the role of the mitochondrial transporters in PDAC initiation and maintenance was unveiled only in the last years and many open issues still require further investigation. We believe some of the issues raised by the reviewer have been considered in several reviews published elsewhere.

https://molecular-cancer.biomedcentral.com/articles/10.1186/s12943-020-01169-7

https://www.sciencedirect.com/science/article/pii/S0923753419459814

https://pubmed.ncbi.nlm.nih.gov/36139512/

https://pubmed.ncbi.nlm.nih.gov/35954462/

https://www.pnas.org/doi/abs/10.1073/pnas.1501605112

Reviewer 2 Report

Metabolic reprogramming, such as Warburg effect and glutamine addiction, is considered as a key hallmark of cancer. In this review paper, Graziantonio Lauria and colleagues summarized the role of mitochondrial solute carriers (glutamine transporter SLC1A5_Var, mitochondrial glutamate carriers, aspartate-glutamate carriers, mitochondrial aspartate transporter and mitochondrial pyruvate carrier) in PDAC metabolic rewiring and progression. Overall, this paper is very comprehensive, and provides a lot of important information about the functions of mitochondrial transporters in PDAC metabolic rewiring. I only have some small concerns about this manuscript.

1) Maybe due to the limited studies of the exact role of mitochondrial transporters in PDAC metabolic rewiring, authors displayed lots of evidence about the role of mitochondrial transporters in the metabolic rewiring of other cancer. In Section 3, authors mainly focus on CRC, and primarily involved in C2C12 and LLC1 cells in Section 4. Authors should find more supporting data in PDAC, or consider changing the title to “emphasize PDAC”, instead of “only focus on PDAC” in the title.

2) Authors need to double-check the latest papers involving the topic of this manuscript. For example, most recent data showed SLC1A5 could enhance proliferative, migrative, and invasive abilities of pancreatic cancer (PC) cells, while it may increase the sensitivity to ferroptosis. (PMID: 35419359).

3) A few careless mistakes were found in the manuscript, e.g.: Line 294 should be “aralar1”, Line 479 should be “it has recently”. Authors need to double-check the whole text.

Author Response

  1. The reviewer is right, in some sections of the manuscript it has been discussed the role of a couple of mitochondrial transporters in other cancers/cell types. This is because the original papers mainly focused their studies on other cancers/cell types, but they are the only available, up to date, in which the role of these transporters (glutamate and glutamate-aspartate carriers) has been also investigated in KRAS mutated PDAC cells. We think that citing these papers and reporting only the data concerning the PDAC is not fair. We also believe it is worthy to cite these studies considering that CRC, likewise PDAC, is also a KRAS dependent tumor and to highlight new research avenues to people working on PDAC. So, we would like to keep the title of manuscript as it is.
  2. Although the manuscript suggested by the reviewer is quite interesting, it focuses on the plasma membrane transporter (SLC1A5) and not on the mitochondrial one (SLC1A_var) arisen from an alternative splicing induced by HIF2a. The role of plasma membrane carriers in pancreatic cancer has been the subject of several reviews cited in our present manuscript (refs 34, 35). We think the data reported on the mitochondrial transporters involved in PDAC are up to date.
  3. Both mistakes have been corrected. The whole text has been double-checked.

Reviewer 3 Report

The article is well written.

I appreciate the coloured schemes.

Author Response

Many thanks